# Tractability in Structured Probability Spaces

**Arthur Choi**
University of California
Los Angeles, CA 90095
aychoi@cs.ucla.edu

**Yujia Shen**
University of California
Los Angeles, CA 90095
yujias@cs.ucla.edu

**Adnan Darwiche**
University of California
Los Angeles, CA 90095
darwiche@cs.ucla.edu

## Abstract

Recently, the Probabilistic Sentential Decision Diagram (PSDD) has been proposed as a framework for systematically inducing and learning distributions over structured objects, including combinatorial objects such as permutations and rankings, paths and matchings on a graph, etc. In this paper, we study the scalability of such models in the context of representing and learning distributions over routes on a map. In particular, we introduce the notion of a hierarchical route distribution and show how they can be leveraged to construct tractable PSDDs over route distributions, allowing them to scale to larger maps. We illustrate the utility of our model empirically, in a route prediction task, showing how accuracy can be increased significantly compared to Markov models.

## 1 Introduction

A structured probability space is one where members of the space correspond to structured or combinatorial objects, such as permutations, partial rankings, or routes on a map [Choi et al., 2015, 2016]. Structured spaces have come into focus recently, given their large number of applications and the lack of systematic methods for inducing and learning distributions over such spaces. Some structured objects are supported by specialized distributions, e.g., the Mallows distribution over permutations [Mallows, 1957, Lu and Boutilier, 2011]. For other types of objects, one is basically on their own as far developing representations and corresponding algorithms for inference and learning. Standard techniques, such as probabilistic graphical models, are not suitable for these kind of distributions since the constraints on such objects often lead to almost fully connected graphical models, which are not amenable to inference or learning.

A framework known as PSDD was proposed recently for systematically inducing and learning distributions over structured objects [Kisa et al., 2014a,b, Shen et al., 2016, Liang et al., 2017]. According to this framework, one first describes members of the space using propositional logic, then compiles these descriptions into Boolean circuits with specific properties (a circuit encodes a structured space by evaluating to 1 precisely on inputs corresponding to members of the space). By parameterizing these Boolean circuits, one can induce a tractable distribution over objects in the structured space. The only domain specific investment in this framework corresponds to the encoding of objects using propositional logic. Moreover, the only computational bottleneck in this framework is the compilation of propositional logic descriptions to circuits with specific properties, which are known as SDD circuits (for Sentential Decision Diagrams) [Darwiche, 2011, Xue et al., 2012]. Parameterized SDD circuits are known as a PSDDs (for Probabilistic SDDs) and have attractive properties, including tractable inference and closed-form parameter estimation under complete data [Kisa et al., 2014a].

Most of the focus on PSDDs has been dedicated to showing how they can systematically induce and learn distributions over various structured objects. Case studies have been reported relating to total and partial rankings [Choi et al., 2015], game traces, and routes on a map [Choi et al., 2016]. The scalability of these studies varied. For partial rankings, experiments have been reported for hundreds

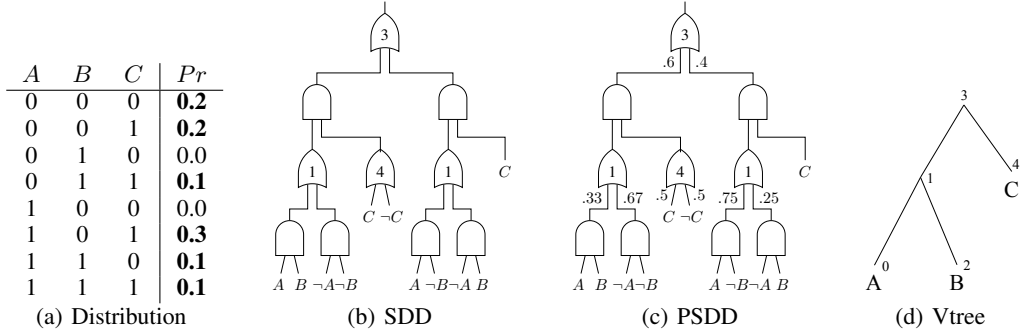

| A | B | C | $Pr$ |
|---|---|---|------|
| 0 | 0 | 0 | **0.2** |
| 0 | 0 | 1 | **0.2** |
| 0 | 1 | 0 | 0.0 |
| 0 | 1 | 1 | **0.1** |
| 1 | 0 | 0 | 0.0 |
| 1 | 0 | 1 | **0.3** |
| 1 | 1 | 0 | **0.1** |
| 1 | 1 | 1 | **0.1** |

(a) Distribution   (b) SDD   (c) PSDD   (d) Vtree

Figure 1: A probability distribution and its SDD/PSDD representation. The numbers annotating or-gates in (b) & (c) correspond to vtree node IDs in (d). While the circuit appears to be a tree, the input variables are shared and hence the circuit is not a tree.

of items. However, for total rankings and routes, the experimental studies were more of a proof of concept, showing for example how the learned PSDD distributions can be superior to ones learned used specialized or baseline methods [Choi et al., 2015].

In this paper, we study a particular structured space, while focusing on computational considerations. The space we consider is that of routes on a map, leading to what we call *route distributions.* These distributions are of great practical importance as they can be used to estimate traffic jams, predict specific routes, and even project the impact of interventions, such as closing certain routes on a map.

The main contribution on this front is the notion of *hierarchical simple-route distributions,* which correspond to a hierarchical map representation that forces routes to be simple (no loops) at different levels of the hierarchy. We show in particular how this advance leads to the notion of hierarchical PSDDs, allowing one to control the size of component PSDDs by introducing more levels of the hierarchy. This guarantees a representation of polynomial size, but at the expense of losing exactness on some route queries. Not only does this advance the state-of-the-art for learning distributions over routes, but it also suggests a technique that can potentially be applied in other contexts as well.

This paper is structured as follows. In Section 2, we review SDD circuits and PSDDs, and in Section 3 we turn to routes as a structured space and their corresponding distributions. Hierarchical distributions are treated in Section 4, with complexity and correctness guarantees. In Section 5, we discuss new techniques for encoding and compiling a PSDD in a hierarchy. We present empirical results in Section 6, and finally conclude with some remarks in Section 7.

## 2 Probabilistic SDDs

PSDDs are a class of tractable probabilistic models, which were originally motivated by the need to represent probability distributions $Pr(\mathbf{X})$ with many instantiations $\mathbf{x}$ attaining zero probability, i.e., a structured space [Kisa et al., 2014a, Choi et al., 2015, 2016]. Consider the distribution $Pr(\mathbf{X})$ in Figure 1(a) for an example. To construct a PSDD for such a distribution, we perform the two following steps. We first construct a special Boolean circuit that captures the zero entries in the following sense; see Figure 1(b). For each instantiation $\mathbf{x}$, the circuit evaluates to $0$ at instantiation $\mathbf{x}$ iff $Pr(\mathbf{x}) = 0$. We then parameterize this Boolean circuit by including a local distribution on the inputs of each or-gate; see Figure 1(c). Such parameters are often learned from data.

The Boolean circuit underlying a PSDD is known as a Sentential Decision Diagram (SDD) [Darwiche, 2011]. These circuits satisfy specific syntactic and semantic properties based on a binary tree, called a *vtree,* whose leaves correspond to variables; see Figure 1(d). SDD circuits alternate between or-gates and and-gates. Their and-gates have two inputs each and satisfy a property called *decomposability*: each input depends on a different set of variables. The or-gates satisfy a property called *determinism*: at most one input will be high under any circuit input. The role of the vtree is (roughly) to determine which variables will appear as inputs for gates.

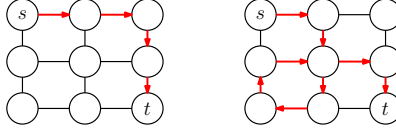

Figure 2: Two paths connecting $s$ and $t$ in a graph.

A PSDD is obtained by including a distribution $\alpha_1, \ldots, \alpha_n$ on the inputs of each or-gate; see again Figure 1(c). The semantics of PSDDs are given in [Kisa et al., 2014a].[1] The PSDD is a complete and canonical representation of probability distributions. That is, PSDDs can represent any distribution, and there is a unique PSDD for that distribution (under some conditions). A variety of probabilistic queries are tractable on PSDDs, including that of computing the probability of a partial variable instantiation and the most likely instantiation. Moreover, the maximum likelihood parameter estimates of a PSDD are unique given complete data, and these parameters can be computed efficiently using closed-form estimates; see [Kisa et al., 2014a] for details. Finally, PSDDs have been used to learn distributions over *combinatorial objects,* including rankings and permutations [Choi et al., 2015], as well as paths and games [Choi et al., 2016]. In these applications, the Boolean circuit underlying a PSDD captures variable instantiations that correspond to combinatorial objects, while its parameterization induces a distribution over these objects.

As a concrete example, PSDDs were used to induce distributions over the permutations of $n$ items as follows. We have a variable $X_{ij}$ for each $i, j \in \{1, \ldots, n\}$ denoting that item $i$ is at position $j$ in the permutation. Clearly, not all instantiations of these variables correspond to (valid) permutations. An SDD circuit is then constructed, which outputs 1 iff the corresponding input corresponds to a valid permutation. Each parameterization of this SDD circuit leads to a distribution on permutations and these parameterizations can be learned from data; see Choi et al. [2015].

## 3   Route Distributions

We consider now the structured space of *simple routes* on a map, which correspond to connected and loop-free paths on a graph. Our ultimate goal here is to learn distributions over simple routes and use them for reasoning about traffic, but we first discuss how to represent such distributions.

Consider a map in the form of an undirected graph $G$ and let $\mathbf{X}$ be a set of binary variables, which are in one-to-one correspondence with the edges of graph $G$. For example, the graph in Figure 2 will lead to 12 binary variables, one for each edge in the graph. A variable instantiation $\mathbf{x}$ will then be interpreted as a set of edges in graph $G$. In particular, instantiation $\mathbf{x}$ includes edge $e$ iff the edge variable is set to true in instantiation $\mathbf{x}$. As such, some of the instantiations $\mathbf{x}$ will correspond to routes in $G$ and others will not.[2] In Figure 2, the left route corresponds to a variable instantiation in which 4 variables are set to true, while all other 8 variables are set to false.

Let $\alpha_G$ be a Boolean formula obtained by disjoining all instantiations $\mathbf{x}$ that correspond to routes in graph $G$. A probability distribution $Pr(\mathbf{X})$ is called a *route distribution* iff it assigns a zero probability to every instantiation $\mathbf{x}$ that does not correspond to a route, i.e., $Pr(\mathbf{x}) = 0$ if $\mathbf{x} \not\models \alpha_G$.

One can systematically induce a route distribution over graph $G$ by simply compiling the Boolean formula $\alpha_G$ into an SDD, and then parameterizing the SDD to obtain a PSDD. This approach was actually proposed in Choi et al. [2016], where empirical results were shown for routes on grids of size at most 8 nodes by 8 nodes.

Let us now turn to simple routes, which are routes that do not contain loops. The path on the left of Figure 2 is simple, while the one on the right is not simple. Among the instantiations $\mathbf{x}$ corresponding to routes, some are *simple routes* and others are not. Let $\beta_G$ be a Boolean formula obtained by disjoining all instantiations $\mathbf{x}$ that correspond to simple routes. We then have $\beta_G \models \alpha_G$.

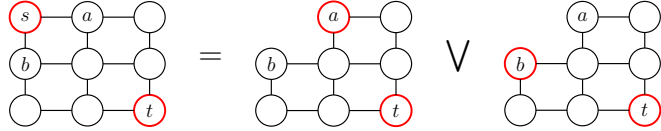

Figure 3: The set of all $s$-$t$ paths corresponds to concatenating edge $(s, a)$ with all $a$-$t$ paths and concatenating edge $(s, b)$ with all $b$-$t$ paths.

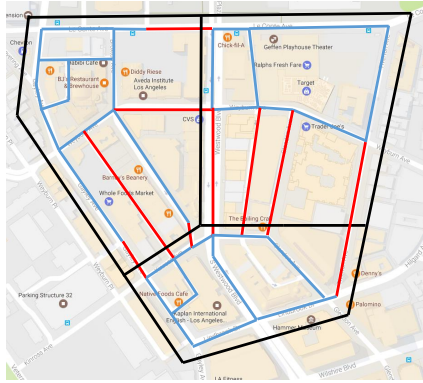

Figure 4: Partitioning a map into three regions (intersections are nodes of the graph and roads between intersections are edges of the graph). Regions have black boundaries. Red edges cross regions and blue edges are contained within a region.

A *simple-route distribution* $Pr(\mathbf{X})$ is a distribution such that $Pr(\mathbf{x}) = 0$ if $\mathbf{x} \not\models \beta_G$. Clearly, simple-route distributions are a subclass of route distributions. One can also systematically represent and learn simple-route distributions using PSDDs. In this case, one must compile the Boolean formula $\beta_G$ into an SDD whose parameters are then learned from data. Figure 3 shows one way to encode this Boolean formula (recursively), as discussed in Choi et al. [2016]. More efficient approaches are known, based on Knuth's Simpath algorithm [Knuth, 2009, Minato, 2013, Nishino et al., 2017].

To give a sense of current scalability when compiling simple-routes into SDD circuits, Nishino et al. [2017] reported results on graphs with as many as 100 nodes and 140 edges for a single source and destination pair. To put these results in perspective, we point out that we are not aware of how one may obtain similar results using standard probabilistic graphical model—for example, a Bayesian or a Markov network. Imposing complex constraints, such as the simple-route constraint, typically lead to highly-connected networks with high treewidths.[3]

While PSDD scalability is favorable in this case—when compared to probabilistic graphical models— our goal is to handle problems that are significantly larger in scale. The classical direction for achieving this goal is to advance current circuit compilation technology, which would allow us to compile propositional logic descriptions that cannot be compiled today. We next propose an alternative, yet a complementary direction, which is based on the notion of hierarchical maps and the corresponding notion of hierarchical distributions.

## 4 Hierarchical Route Distributions

A route distribution can be represented hierarchically if one imposes a hierarchy on the underlying map, leading to a representation that is polynomial in size if one includes enough levels in the hierarchy. Under some conditions which we discuss later, the hierarchical representation can also support inference in time polynomial in its size. The penalty incurred due to this hierarchical representation is a loss of exactness on some queries, which can be controlled as we discuss later.

We start by discussing *hierarchical maps,* where a map is represented by a graph $G$ as discussed earlier. Let $\mathbf{N}_1, \ldots, \mathbf{N}_m$ be a partitioning of the nodes in graph $G$ and let us call each $\mathbf{N}_i$ a *region.* These regions partition edges $\mathbf{X}$ into $\mathbf{B}, \mathbf{A}_1, \ldots, \mathbf{A}_m$, where $\mathbf{B}$ are edges that cross regions and $\mathbf{A}_i$ are edges inside region $\mathbf{N}_i$. Consider the following decomposition for distributions over routes:

$$Pr(\mathbf{x}) = Pr(\mathbf{b}) \prod_{i=1}^{m} Pr(\mathbf{a}_i \mid \mathbf{b}_i). \tag{1}$$

We refer to such a distribution as a *decomposable route distribution.*[4] Here, $\mathbf{B}_i$ are edges that cross out of region $\mathbf{N}_i$, and $\mathbf{b}$, $\mathbf{a}_i$ and $\mathbf{b}_i$ are partial instantiations that are compatible with instantiation $\mathbf{x}$.

To discuss the main insight behind this hierarchical representation, we need to first define a graph $G_{\mathbf{B}}$ that is obtained from $G$ by aggregating each region $\mathbf{N}_i$ into a single node. We also need to define subgraphs $G_{\mathbf{b}_i}$, obtained from $G$ by keeping only edges $\mathbf{A}_i$ and the edges set positively in instantiation $\mathbf{b}_i$ (the positive edges of $\mathbf{b}_i$ denote the edges used to enter and exit the region $\mathbf{N}_i$).

Hence, graph $G_{\mathbf{B}}$ is an abstraction of graph $G$, while each graph $G_{\mathbf{b}_i}$ is a subset of $G$. Moreover, one can think of each subgraph $G_{\mathbf{b}_i}$ as a local map (for region $i$) together with a particular set of edges that connects it to other regions. We can now state the following key observations. The distribution $Pr(\mathbf{B})$ is a route distribution for the aggregated graph $G_{\mathbf{B}}$. Moreover, each distribution $Pr(\mathbf{A}_i \mid \mathbf{b}_i)$ is a distribution over (sets of) routes for subgraph $G_{\mathbf{b}_i}$ (in general, we may enter and exit a region multiple times).

Hence, we are able to represent the route distribution $Pr(\mathbf{X})$ using a set of smaller route distributions. One of these distributions $\Pr(\mathbf{B})$ captures routes across regions. The others, $Pr(\mathbf{A}_i \mid \mathbf{b}_i)$, capture routes that are within a region. The count of these smaller distributions is $1 + \sum_{i=1}^{m} 2^{|\mathbf{B}_i|}$, which is exponential in the size of variable sets $\mathbf{B}_1, \ldots, \mathbf{B}_n$. We will later see that this count can be polynomial for some simple-route distributions.

We used $\alpha_G$ to represent the instantiations corresponding to routes, and $\beta_G$ to represent the instantiations corresponding to simple routes, with $\beta_G \models \alpha_G$. Some of these simple routes are also simple with respect to the aggregated graph $G_{\mathbf{B}}$ (i.e., they will not visit a region $\mathbf{N}_i$ more than once), while other simple routes are not simple with respect to graph $G_{\mathbf{B}}$. Let $\gamma_G$ be the Boolean expression obtained by disjoining instantiations $\mathbf{x}$ that correspond to simple routes that are also simple (and non-empty) with respect to graph $G_{\mathbf{B}}$.[5] We then have $\gamma_G \models \beta_G \models \alpha_G$ and the following result.

**Theorem 1** *Consider graphs $G$, $G_{\mathbf{B}}$ and $G_{\mathbf{b}_i}$ as indicated above. Let $Pr(\mathbf{B})$ be a simple-route distribution for graph $G_{\mathbf{B}}$, and $Pr(\mathbf{A}_i \mid \mathbf{b}_i)$ be a simple-route distribution for graph $G_{\mathbf{b}_i}$. Then the resulting distribution $Pr(\mathbf{X})$, as defined by Equation 1, is a simple-route distribution for graph $G$.*

This theorem will not hold if $Pr(\mathbf{B})$ were not a simple-route distribution for graph $G_{\mathbf{B}}$. That is, having each distribution $Pr(\mathbf{A}_i \mid \mathbf{b}_i)$ be a simple-route distribution for graph $G_{\mathbf{b}_i}$ is not sufficient for the hierarchical distribution to be a simple-route distribution for $G$.

Hierarchical distributions that satisfy the conditions of Theorem 1 will be called *hierarchical simple-route distributions.*

**Theorem 2** *Let $Pr(\mathbf{X})$ be a hierarchical simple-route distribution for graph $G$ and let $\gamma_G$ be as indicated above. We then have $Pr(\mathbf{x}) = 0$ if $\mathbf{x} \not\models \gamma_G$.*

This means that the distribution will assign a zero probability to all instantiations $\mathbf{x} \models \beta_G \wedge \neg\gamma_G$. These instantiations correspond to routes that are simple for graph $G$ but not simple for graph $G_{\mathbf{B}}$. Hence, simple-route hierarchical distributions correspond to a subclass of the simple-route distributions for graph $G$. This subclass, however, is interesting for the following reason.

**Theorem 3** *Consider a hierarchical simple-route distribution $Pr(\mathbf{X})$ and let $\mathbf{x}$ be an instantiation that sets more than two variables in some $\mathbf{B}_i$ to true. Then $Pr(\mathbf{x}) = 0$.*

Basically, a route that is simple for graph $G_\mathbf{B}$ cannot enter and leave a region more than once.

**Corollary 1** *The hierarchical simple-route distribution $Pr(\mathbf{X})$ can be constructed from distribution $Pr(\mathbf{B})$ and distributions $Pr(\mathbf{A}_i \mid \mathbf{b}_i)$ for which $\mathbf{b}_i$ sets no more than two variables to true.*

**Corollary 2** *The hierarchical simple-route distribution $Pr(\mathbf{X})$ can be represented by a data structure whose size is $O(2^{|\mathbf{B}|} + \sum_{i=1}^{m} 2^{|\mathbf{A}_i|}|\mathbf{B}_i|^2)$.*

If we choose our regions $\mathbf{N}_i$ to be small enough, then $2^{|\mathbf{A}_i|}$ can be treated as a constant. A tabular representation of the simple-route distribution $Pr(\mathbf{B})$ has size $O(2^{|\mathbf{B}|})$. If representing this table is practical, then inference is also tractable (via variable elimination). However, this distribution can itself be represented by a simple-route hierarchical distribution. This process can continue until we reach a simple-route distribution that admits an efficient representation. We can therefore obtain a final representation which is polynomial in the number of variables $\mathbf{X}$ and, hence, polynomial in the size of graph $G$ (however, inference may no longer be tractable).

In our approach, we represent the distributions $Pr(\mathbf{B})$ and $Pr(\mathbf{A}_i \mid \mathbf{b}_i)$ using PSDDs. This allows these distributions to be over a relatively large number of variables (on the order of hundreds), which would not be feasible if we used more classical representations, such as graphical models.

This hierarchical representation, which is both small and admits polytime inference, is an approximation as shown by the following theorem.

**Theorem 4** *Consider a decomposable route distribution $Pr(\mathbf{X})$ (as in Equation 1), the corresponding hierarchical simple-route distribution $Pr(\mathbf{X} \mid \gamma_G)$, and a query $\alpha$ over variables $\mathbf{X}$. The error of the query $Pr(\alpha \mid \gamma_G)$, relative to $Pr(\alpha)$, is:*

$$\frac{Pr(\alpha \mid \gamma_G) - Pr(\alpha)}{Pr(\alpha \mid \gamma_G)} = Pr(\kappa_G)\left[1 - \frac{Pr(\alpha \mid \kappa_G)}{Pr(\alpha \mid \gamma_G)}\right]$$

*where $\kappa_G = \beta_G \wedge \neg\gamma_G$ denotes simple-routes in $G$ that are not simple routes in $G_\mathbf{B}$.*

The conditions of this theorem basically require the two distributions to agree on the relative probabilities of simple routes that are also simple in $G_\mathbf{B}$. Note also that $Pr(\gamma_G) + Pr(\kappa_G) = 1$. Hence, if $Pr(\gamma_G) \approx 1$, then we expect the hierarchical distribution to be accurate. This happens when most simple routes are also simple in $G_\mathbf{B}$, a condition that may be met by a careful choice of map regions.[6] At one extreme, if each region has at most two edges crossing out of it, then $Pr(\gamma_G) = 1$ and the hierarchical distribution is exact.

Hierarchical simple-route distributions will assign a zero probability to routes $\mathbf{x}$ that are simple in $G$ but not in $G_\mathbf{B}$. However, for a mild condition on the hierarchy, we can guarantee that if there is a simple route between nodes $s$ and $t$ in $G$, there is also a simple route that is simple for $G_\mathbf{B}$.

**Proposition 1** *If the subgraphs $G_{\mathbf{b}_i}$ are connected, then there is a simple route connecting $s$ and $t$ in $G$ iff there is a simple route connecting $s$ and $t$ in $G$ that is also a simple route for $G_\mathbf{B}$.*

Under this condition, hierarchical simple-route distributions will provide an approximation for any source/destination query.

One can compute marginal and MAP queries in polytime on a hierarchical distribution, assuming that one can (in polytime) multiply and sum-out variables from its component distributions—we basically need to sum-out variables $\mathbf{B}_i$ from each $Pr(\mathbf{A}_i|\mathbf{b}_i)$, then multiply the results with $Pr(\mathbf{B})$. In our experiments, however, we follow a more direct approach to inference, in which we multiply all component distributions (PSDDs), to yield one PSDD for the hierarchical distribution. This is not always guaranteed to be efficient, but leads to a much simpler implementation.

## 5 Encoding and Compiling Routes

Recall that constructing a PSDD involves two steps: constructing an SDD that represents the structured space, and then parameterizing the SDD. In this section, we discuss how to construct

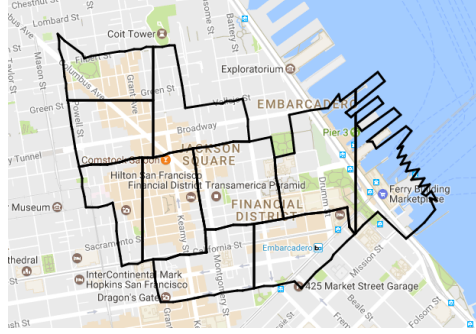

Figure 5: Partitioning of the area around the Financial District of San Francisco, into regions.

an SDD that represents the structured space of hierarchical, simple routes. Subsequently, in our experiments, we shall learn the parameters of the PSDD from data.

We first consider the space of simple routes that are not necessarily hierarchical. Note here that an SDD of a Boolean formula can be constructed bottom-up, starting with elementary SDDs representing literals and constants, and then constructing more complex SDDs from them using conjoin, disjoin, and negation operators implemented by an SDD library. This approach can be used to construct an SDD that encodes simple routes, using the idea from Figure 3, which is discussed in more detail in Choi et al. [2016]. The GRAPHILLION library can be used to construct a Zero-suppressed Decision Diagram (ZDD) representing all simple routes for a given source/destination pair [Inoue et al., 2014]. The ZDDs can then be disjoined across all source and destination pairs, and then converted to an SDD. An even more efficient algorithm was proposed recently for compiling simple routes to ZSDDs, which we used in our experiments [Nishino et al., 2016, 2017].

Consider now the space of hierarchical simple routes induced by regions $\mathbf{N}_1, \ldots, \mathbf{N}_m$ of graph $G$, with a corresponding partition of edges into $\mathbf{B}, \mathbf{A}_1, \ldots, \mathbf{A}_m$, as discussed earlier. To compile an SDD for the hierarchical, simple routes of $G$, we first compile an SDD representing the simple routes over each region. That is, for each region $\mathbf{N}_i$, we take the graph induced by the edges $\mathbf{A}_i$ and $\mathbf{B}_i$, and compile an SDD representing all its simple routes (as described above). Similarly, we compile an SDD representing the simple routes of the abstracted graph $G_{\mathbf{B}}$. At this point, we have a hierarchical, simple-route distribution in which components are represented as PSDDs and that we can do inference on using multiplication and summing-out as discussed earlier.

In our experiments, however, we take the extra step of multiplying all the $m + 1$ component PSDDs, to yield a single PSDD over the structured space of hierarchical, simple routes. This simplifies inference and learning as we can now use the linear-time inference and learning procedures known for PSDDs [Kisa et al., 2014a].[7]

## 6 Experimental Results

In our experiments, we considered a dataset consisting of GPS data collected from taxicab routes in San Francisco.[8] We acquired public map data from `http://www.openstreetmap.org/`, i.e., the undirected graph representing the streets (edges) and intersections (nodes) of San Francisco. We projected the GPS data onto the San Francisco graph using the map-matching API of the `graphhopper` package.[9] For more on map-matching, see, e.g., [Froehlich and Krumm, 2008].

To partition the graph of San Francisco into regions, we obtained a publicly available dataset of traffic analysis zones, produced by the California Metropolitan Transportation Commission.[10] These zones correspond to small area neighborhoods and communities of the San Francisco Bay Area. To facilitate the compilation of regions into SDDs, we further split these zones in half until each region was compilable (horizontally if the region was taller than it was wide, or vertically otherwise). Finally, we restricted our attention to areas around the Financial District of San Francisco, which we were able to compile into a hierarchical distribution using one level of abstraction; see Figure 5.

Given the routes over the graph of San Francisco, we first filtered out any routes that did not correspond to a simple path on the San Francisco graph. We next took all routes that were contained solely in the region under consideration. We further took any sub-route that passed through this region, as a route for our region. In total, we were left with 87,032 simple routes. We used half for training, and the other half for testing. For the training set, we also removed all simple routes that were not simple in the hierarchy. We did not remove such routes for the purposes of testing. We first compiled an SDD of hierarchical simple-routes over the region, leading to an SDD with 62,933 nodes, and 152,140 free parameters. We then learned the parameters of our PSDD from the training set, assuming Laplace smoothing [Kisa et al., 2014a].

We considered a route prediction task where we predict the next road segment, given the route taken so far; see, e.g., [Letchner et al., 2006, Simmons et al., 2006, Krumm, 2008]. That is, for each route of the testing set, we consider one edge at a time and try to predict the next edge, given the edges observed so far. We consider three approaches: (1) a naive baseline that uses the relative frequency of edges to predict the next edge, while discounting the last-used edge, (2) a Markov model that predicts, given the last-used edge, what edge would be the most likely one to be traversed next, (3) a PSDD given the current partial route as well as the destination. The last assumption is often the situation in reality, given the ubiquity of GPS routing applications on mobile phones. We remark that Markov models and HMMs are less amenable to accepting a destination as an observation.

For the PSDD, the current partial route and the last edge to be used (i.e., the destination) are given as evidence $\mathbf{e}$. The evidence for an endpoint (source or destination) is the edge used (set positively), where the remaining edges are assumed to be unused (and set negatively). For internal nodes on a route, two edges (entering and exiting a node) are set positively and the remaining edges are set negatively in the evidence. To predict the next edge on a partial route, we consider the edges $X$ incident to the current node and compute their marginal probabilities $Pr(X \mid \mathbf{e})$ according to the PSDD. The probability of the last edge used in the partial route is 1, which we ignore. The remaining edges have a probability that sums to a value less than one; one minus this probability is the probability that the route ends at the current node. Among all these options, we pick the most likely as our prediction (either navigate to a new edge, or stop).

Note that for the purposes of training our PSDD, we removed those simple routes that were not simple on the hierarchy. When testing, such routes have a probability of zero on our PSDD. Moreover, partial routes may also have zero probability, if they cannot be extended to a hierarchical simple-route. In this case, we cannot compute the marginals $Pr(X \mid \mathbf{e})$. Hence, we simply unset our evidence, one edge at a time in the order that we set them (first unsetting negative edges before positive edges), until the evidence becomes consistent again, relative to the PSDD.

We summarize the relative accuracies over 43,516 total testing routes:

| model | naive | Markov | PSDD |
|---|---|---|---|
| accuracy | 0.736 (326,388/443,481) | 0.820 (363,536/443,481) | 0.931 (412,958/443,481) |

For each model, we report the accuracy averaged over all steps on all paths, ignoring those steps where the prediction is trivial (i.e., there is only one edge or no edge available to be used next). We find that the PSDD is much more accurate at predicting the next road segment, compared to the Markov model and the naive baseline. Indeed, this could be expected as (1) the PSDD uses the history of the route so far, and perhaps more importantly, (2) it utilizes knowledge of the destination.

# 7 Conclusion

In this paper, we considered Probabilistic Sentential Decision Diagrams (PSDDs) representing distributions over routes on a map, or equivalently, simple paths on a graph. We considered a hierarchical approximation of simple-route distributions, and examined its relative tractability and its accuracy. We showed how this perspective can be leveraged to represent and learn more scalable PSDDs for simple-route distributions. In a route prediction task, we showed that PSDDs can take advantage of the available observations, such as the route taken so far and the destination of a trip, to make more accurate predictions.

### Acknowledgments

We greatly thank Noah Hadfield-Menell and Andy Shih for their contributions, and Eunice Chen for helpful discussions. This work has been partially supported by NSF grant #IIS-1514253, ONR grant #N00014-15-1-2339 and DARPA XAI grant #N66001-17-2-4032.

## Footnotes

[1]Let $\mathbf{x}$ be an instantiation of PSDD variables. If the SDD circuit outputs 0 at input $\mathbf{x}$, then $Pr(\mathbf{x}) = 0$. Otherwise, traverse the circuit top-down, visiting the (unique) high input of each visited or-node, and all inputs of each visited and-node. Then $Pr(\mathbf{x})$ is the product of parameters visited during the traversal process.

[2]An instantiation $\mathbf{x}$ corresponds to a route iff the edges it mentions positively can be ordered as a sequence $(n_1, n_2), (n_2, n_3), (n_3, n_4), \ldots, (n_{k-1}, n_k)$.

[3]If we can represent a uniform distribution of simple routes on a map, then we can count the number of simple paths on a graph, which is a #P-complete problem [Valiant, 1979]. Hence, we do not in general expect a Bayesian or Markov network for such a distribution to have bounded treewidth.

[4]Note that not all route distributions can be decomposed as such: the decomposition implies the independence of routes on edges $\mathbf{A}_i$ given the route on edges $\mathbf{B}$.

[5]For most practical cases, the independence assumption of the hierarchical decomposition will dictate that routes on $G_{\mathbf{B}}$ be non-empty. An empty route on $G_{\mathbf{B}}$ corresponds to a route contained within a single region, which we can accommodate using a route distribution for the single region.

[6] If $\alpha$ is independent of $\gamma_G$ (and hence $\alpha$ is independent of $\kappa_G$), then the approximation is also exact. At this point, however, we do not know of an intuitive characterization of queries $\alpha$ that satisfy this property.

[7]In our experiments, we use an additional simplification. Recall from Footnote 5 that if $\mathbf{b}_i$ sets all variables negatively (i.e., no edges), then $G_{\mathbf{b}_i}$ is empty. We now allow the case that $G_{\mathbf{b}_i}$ contains all edges $\mathbf{A}_i$ (by disjoing the corresponding SDDs). Intuitively, this optionally allows a simple path to exist strictly in region $\mathbf{R}_i$. While the global SDD no longer strictly represents hierarchical simple paths (it may allow sets of independent simple paths at once), we do not have to treat simple paths that are confined to a single region as a special case.

[8]Available at `http://crawdad.org/epfl/mobility/20090224/`.

[9]Available at `https://www.graphhopper.com`.

[10]Available at `https://purl.stanford.edu/fv911pc4805`.

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
