[Reviews · NeurIPS 2017]

Reviewer 1



This paper looks at the problem of representing simple routes on a graph as a probability distribution using Probabilistic Sentential Decision Diagrams (PSDDs). Representing a complex structure such as a graph is difficult, and the authors transform the problem by turning a graph into a Boolean circuit where it is straightforward to perform inference, and as an experiment, use their method on a route prediction method for San Francisco taxi cabs, where it beats two baselines. PSDDs refer to a framework that represents probability distributions over structured objects through Boolean circuits. Once the object is depicted as a Boolean circuit, it becomes straightforward to parameterize it. More formally, PSDD’s are parameterized by including a distribution over each or-gate, and PSDD’s can represent any distribution (and under some conditions, this distribution is unique). The authors focus on the specific problem of learning distributions over simple routes — those that are connected and without cycles. The advantage of SDD circuits is that they have been shown to work on graphs that would be computationally expensive to model with Bayesian nets. With large maps, tractability is still a problem with PSDDs, and the authors do this by considering hierarchical approximations; they break up the map into hierarchies so that representing distributions is polynomial when constraining the size of each region to be a certain size. The paper is well-written, and the authors define PSDD's clearly and succinctly. To me, the results of the paper seems incremental -- the authors apply an existing representation method to graphs, and after some approximations, apply the existing inference techniques. Additionally, the paper spends a couple of pages listing theorems, which appear to be basic algorithmic results. Additionally, I'm not convinced by the baselines chosen for the experiment. For a task of predicting edges given a set of previous edges, the baselines are a naive model which only looks at frequencies and a Markov model that only considers the last edge used. I imagine a deep learning approach or even a simple probabilistic diagram could've yielded more interesting and comparable results. Even if these would require more time to train, it would be valuable to compare model complexity in addition to accuracy. I would need to see more experiments to be convinced by the approach.

Reviewer 2



The authors develop a hierarchical version of a probabilistic sequential decision diagrams (PSDD) in the context of describing distributions over routes in a discrete graph, which are represented by binary random variables on the graph edges. Because valid paths without loops are represented as functions of all the edges in a path, a naive graphical model representation would have high treewidth. PSDDs can represent distributions on structured binary spaces more efficiently, but still have difficulty scaling to realistic situations. The authors propose segmenting the problem in a hierarchy of problems, represented both sub-problems and the overall structure by separate PSDDs, allowing the authors to scale to real-world sized problems (e.g. the financial district of San Francisco). The authors rely on some special structure of simple path-finding to make their approach tractable (e.g. the simple-route limitation which 2^|B_i| to |B_i|^2). Thm 4 and Prop 1 provide some guidance to the validity of the underlying assumptions, but in practice they may be hard to evaluate on really difficult problems (as always, I guess). I feel like the Markov model is not a very fair comparison -- as the author say, it does not incorporate destination information. A comparison to route-planning heuristics that are used in practice would have been more appropriate, but I'm afraid I'm not familiar enough with the field to make concrete suggestions. Despite these minor shortcomings, I think the paper is well-written and the idea interesting enough to be of general interest, and I recommend acceptance.

Reviewer 3



The paper introduced hierarchical Probabilistic Sentential Decision Diagram and studied its performance in route prediction. It shows how introducing the hierarchical idea preserves the properties of SDD and the improvement in route prediction. It also presents why this hierarchical PSDD enjoys more efficient computation compared to probabilistic graphical models -- due to its efficiency in describing feasible combinatorial structures. The paper is overall quite nice except it only presented an illustration of hierarchical PSDD on one example dataset/task. Would be helpful to see if this advantage persists through other datasets. Another question is how computation cost of hierarchical PSDD compares of the original PSDD. Given a more flexible model (hierarhical), the computation cost would probably be higher -- but how much higher is interesting to know. ---------------------- After author rebuttal: Many thanks for the rebuttal from the authors. They do clarify. My scores remain the same though. Thanks!